# Growth Trajectories during the First 6 Years in Survivors Born at Less Than 25 Weeks of Gestation Compared with Those between 25 and 29 Weeks

**DOI:** 10.3390/jcm11051418

**Published:** 2022-03-04

**Authors:** Hiromichi Shoji, Yayoi Murano, Shuko Nojiri, Yoshiteru Arai, Kentaro Awata, Naho Ikeda, Natsuki Ohkawa, Naoto Nishizaki, Hiroki Suganuma, Ken Hisata, Masato Kantake, Kaoru Obinata, Toshiaki Shimizu

**Affiliations:** 1Department of Pediatrics, Faculty of Medicine, Juntendo University, Tokyo 113-8421, Japan; ymurano@juntendo.ac.jp (Y.M.); yo-arai@juntendo.ac.jp (Y.A.); k-awata@juntendo.ac.jp (K.A.); hsuganu@juntendo.ac.jp (H.S.); ken-hisa@juntendo.ac.jp (K.H.); tshimizu@juntendo.ac.jp (T.S.); 2Clinical Translational Science, Juntendo University Graduate School of Medicine, Tokyo 113-8421, Japan; s-nojiri@juntendo.ac.jp; 3Department of Neonatology, Juntendo University Shizuoka Hospital, 1129 Nagaoka, Izunokuni-shi, Shizuoka 410-2295, Japan; nikeda@juntendo.ac.jp (N.I.); nookawa@juntendo.ac.jp (N.O.); 4Department of Pediatrics, Juntendo University Urayasu Hospital, Chiba 279-0021, Japan; nishizak@juntendo.ac.jp (N.N.); kobinata@juntendo.ac.jp (K.O.); 5Department of Neonatology, Juntendo University Nerima Hospital, Tokyo 136-0075, Japan; kantake@juntendo.ac.jp

**Keywords:** preterm infants, periviable birth, body mass index, anthropometric z-scores

## Abstract

We aimed to determine the differences in the growth trajectories of the youngest gestational survivors (<25 weeks’ gestation) up to 6 years of age compared to those of older gestational ages. Preterm infants were divided into two groups: 22–24 weeks’ gestation (male (M) 16, female (F) 28) and 25–29 weeks’ gestation (M 84, F 59). Z-scores of body weight (BW), body length (BL), and body mass index (BMI) were derived from Japanese standards at 1, 1.5, 3, and 6 years of corrected age. Comparisons between the two groups by sex were made using the Wilcoxon test and linear regression analysis to examine the longitudinal and time-point associations of anthropometric z-scores, the presence of small for gestational age (SGA), and the two gestational groups. BW, BL, BMI, and z-scores were significantly lower in the 22–24 weeks group at almost all assessment points. However, there were no significant differences in BW, BL, BMI, and z-scores between the two female groups after 3 years. BMI z-scores were significantly associated with the youngest gestational age and the presence of SGA at all ages in males, but not in females. The youngest gestational age had a greater influence in males on the z-score of anthropometric parameters up to 6 years of age.

## 1. Introduction

Recent progression in neonatal intensive care has led to dramatic improvements in the survival of very low birth weight (VLBW, birth weight < 1500 g) infants and very preterm infants [1]. Circulatory management guided by neonatologist-performed echocardiography and nutritional management promoted by early enteral feeding contributed to decreasing the incidence of intraventricular hemorrhage (IVH) and necrotizing enterocolitis (NEC) in Japan [2]. However, the care of periviable births (defined as delivery between 20 and 25 weeks of gestation) remains a great challenge in neonatal and perinatal medicine [3]. Infants born at <25 weeks of gestation have high mortality rates and experience long-term neurodevelopmental impairments [4,5,6]. Anderson et al. recently investigated a total of 1818 infants born at <25 weeks and reported that survival rates after resuscitation were 54.6%, and 83% of deaths occurred in the first week of life [7]. In Japan, mortality in neonates born at <25 weeks of gestation is remarkably low compared to that in other countries [8]. Isayama et al. recently reported a 27.1% mortality rate in infants at <25 weeks’ gestation [9]. Such studies raise questions about growth outcomes in surviving periviable births infants. Although nutritional strategies are advancing, including higher intravenous amino acids, lipid administration, and the earlier achievement of full enteral feeding, postnatal (extreuterine) growth failure in VLBW infants remains a significant problem [10]. Takayanagi et al. reported that the prevalence of extrauterine growth restriction was 47.5% among 322 VLBW infants, and it was significantly associated with short stature and thinness at 6 years [11]. However, there are limited longitudinal studies examining the trajectory of anthropometric parameters in children of periviable birth in Japan. The aim of the present study was to determine changes in the growth trajectories of the youngest gestational survivors (<25 weeks’ gestation) up to 6 years of age compared with those of older gestational ages.

## 2. Materials and Methods

A retrospective multicenter cohort study was conducted that including preterm infants admitted to neonatal intensive care units (NICUs) in the three affiliated hospitals of Juntendo University (Juntendo University Hospital in Tokyo, Juntendo University Urayasu Hospital in Chiba, and Juntendo University Shizuoka Hospital in Shizuoka) from January 2007 to June 2012. The inclusion criteria for preterm infants were birth weight <1500 g, gestational age (GA) <30 weeks and Japanese parentage. Participants were divided into two subgroups: 22–24 weeks of gestation (22–24 weeks) and 25–29 weeks of gestation (25–29 weeks). GA was determined by last menstrual period and first-trimester ultrasonogram. Body weight (BW) and body length (BL) at birth and 1, 1.5, 3, and 6 years of corrected age (CA, representing the age of the children from the expected date of delivery) were collected from medical records, and the Ponderal index (g/cm^3^) or body mass index (BMI; kg/m^2^) was calculated. Sex- and GA-independent z-scores and percentiles for anthropometric parameters at birth were calculated according to the Japanese standard charts, which were estimated using 2003–2005 data from the Japan Society of Obstetrics and Gynecology registry database [12]. SGA was defined as birth weight for a GA < 10 percentile.

The exclusion criteria were infants with the presence of congenital diseases, chromosomal abnormalities, or severe cardiac, renal, or endocrine diseases. Infants whose birth weight z-scores were >2 and children who received growth hormone therapy were excluded. We collected the data of several clinical complications.

Z-scores of BW, BL, and BMI at 1 1.5, 3, and 6 years of CA were also calculated according to the standard growth chart for children from a national survey conducted in 2000 [13]. Excel-based software was developed for both standard curves by the Japanese Society for Pediatric Endocrinology; this software is available on their website [14].

A research investigator and research team members in each hospital gained approval for this study from the Ethics Committee of Juntendo University Hospital (17-094), Juntendo University Shizuoka Hospital (2017-059) and Juntendo University Urayasu Hospital (rin-541). All methods in this retrospective study were performed in accordance with the relevant guidelines (Ethical Guidelines for Medical and Health Research Involving Human Subjects). Because this is a retrospective study, it was very difficult to gain appropriate informed consents from each subject. Therefore, we gained consent using opt-out, which is a way for investigators to give subjects and their guardians an opportunity to refuse to participate in this study by announcing the details of the study in each participating hospital.

Data are presented as the mean and standard deviation. Differences between the groups were identified using the Wilcoxon test. These statistical analyses were conducted using JMP statistical software (version 12.0; SAS Institute Inc., Cary, NC, USA). Statistical significance was set at *p* < 0.05. The anthropometric parameters were individually plotted between 1 and 6 years of CA to visually inspect the longitudinal pattern. We performed linear regression analysis to determine the association between anthropometric z-scores and the presence of SGA and GA. First, the samples were divided into two GA groups. One group consisted of infants born between 22 and 24 GA, and the other group was born between 25 and 29 GA. We set the group born between 22 and 24 GA and infants with SGA as the reference category and performed linear regression analysis with these variables as independent variables. Dependent variables were z-scores of BW, BL, and BMI at 1, 1.5, 3, and 6 years of CA. Regression analysis was performed separately by sex. Data were analyzed using SAS version 9.4 (SAS Institute, Inc., Cary, NC, USA).

## 3. Results

During the study period, a total of 529 VLBW infants with GA < 30 weeks were born in the NICUs of three hospitals, and 502 infants met the inclusion criteria at birth. Of these infants, 412 patients were discharged from the NICUs. We could collect the data from 197 children up to 6 years of CA. We excluded ten children who received growth hormone therapy. We analyzed 187 remaining children, of whom 44 were categorized into the 22–24 weeks group (Male (M) 16, Female (F) 28), and 143 in the 25–29 weeks group (M 84, F 59) (Figure 1).

Table 1 presents the clinical characteristics and anthropometric indices of the participants. GA, birth weight, BL, and Ponderal index at birth were significantly lower in the 22–24 weeks group than in the 25–29 weeks group for both sexes (*p* < 0.05). In females, the z-score of birth weight and BL at birth were also significantly lower in the 22–24 weeks group than those in the 25–29 weeks group, but not in males. BW, BL, and BMI and their z-scores were significantly lower in the 22–24 weeks group than those in the 25–29 weeks group for all assessments. However, there were no significant differences in BW, BL, and BMI and their z-scores between the two female groups in all assessments, but there was a significant difference in BW z-score at 1.5 years of CA.

For males, the z-scores of BW, BL, and BMI were significantly lower in the 22–24 weeks group than those in the 25–29 weeks group at all time points except for the BL z-score at CA of 3 years. For females, no significant differences in the z-scores of BW, BL, or BMI were seen between the 22–24 and 25–29 weeks groups at all time points except for BW z-score at CA 1.5 years (Figure 2).

Linear regression analysis showed that the association between BW z-scores and both GA and the presence of SGA was seen in all age groups in both sexes, except for 6 years of CA in females (Table 2). The association between BL z-scores and both GA groups and the presence of SGA was observed in males at all time points (Table 3). However, in females, the association between BL z-scores at 1.5 years of CA and GA, BL z-scores at 3 years of CA and GA, BL z-scores at 6 years of CA and both GA groups, and the presence of SGA were not observed in our analysis (Table 3). BMI z-scores were significantly associated with GA and the presence of SGA at all time points in males, but this association was not observed in females (Table 4).

## 4. Discussion

This is, to our best knowledge, the first study to assess the influence of younger gestational age (<25 weeks’ gestation) on anthropometric indices using z-scores during the first 6 years of life among Japanese VLBW infants born at <30 weeks of gestation. We limited preterm infants by GA rather than birth weight (e.g., <1500 g) to decrease bias, because more mature children with FGR were included. Our previous study demonstrated that FGR affected the z-scores of BW and BMI in males and of BW and BL in females during 6 years among same population assessed by ANOVA [15]. However, our following study showed the importance of GA in anthropometric parameters using a larger cohort of 8838 eligible samples [16]. Another study reported that FGR affected anthropometric parameters among children in preterm infants [17]. Therefore, we analyzed both FGR and GA in this study. We found that FGR (nearly equal to SGA) was associated with anthropometric z-scores at 6 years of CA in males but not in females, as assessed by linear regression analysis.

A recent systematic review indicated that accelerated weight gain (defined as weight gain velocity during the first two years) significantly increased childhood obesity among preterm infants [18]. Several studies have reported increased rates of obesity and being overweight among preterm children in adolescence and early adulthood [19,20]. Although there were no sex differences among these studies, hierarchical analyses for GA and sex are needed to clarify the risk of obesity in VLBW infants. Our longitudinal study demonstrated that z-scores of BW, BL, and BMI in females of the 22–24 weeks group were close to those in the 25–29 weeks group, but in males, they were parallel during the first 6 years.

Poor growth attainment in early childhood and early adolescence was reported in extremely preterm survivors [20,21,22]. In this study, children born at <25 weeks were 1.5 z-scores lighter and 1.0 z-score shorter in males, and 1.1 z-scores lighter and 0.7 z-scores shorter in females than in the controls at 6 years of CA. The EPICure study, a large prospective, population-based cohort study compared children born at <26 weeks with term controls and reported that preterm children were 1.2 SD lighter and 0.97 SD shorter than term control at 6 years [20]. Kytnarova et al. also reported that BW and BL were 1.0 SD lower than children born at <26 weeks and 0.7 SD lower for children born at 26–28 weeks [23].

The previously reported fat-free mass index was lower in males born preterm than that in females, whereas no difference in BMI was detected in either sex [24]. Persistent deficits in fat-free mass at 5 years of age exhibited by male born preterm infants may be explained by the exposure to nutritional, hormonal, and environmental influences during hospital stay, which may have affected the preferred partitioning of nutrients to lean tissue growth [25]. A study also reported catch-up weight among VLBW females in young adulthood, but not among VLBW males [26]. The authors argue that this might be related to the lower neonatal and early childhood morbidity among VLBW females [26]. Although we found differences between the sexes in growth parameters, the incidence of IVH, PDA, NEC, CLD, PVL, and ROP were not significantly different between the sexes in this study. Further study is required to clarify sex differences in anthropometric indices among Japanese preterm children.

This study has several limitations. The follow-up rate of participants was low (45%); we analyzed the characteristics of infants with and without follow-up (Appendix A); and female infants aged 25–29 weeks who dropped out of this study had larger anthropometric parameters at birth and a lower prevalence of SGA. Although anthropometric measurements at birth in infants with 22–24 weeks gestation did not differ between the follow-up group and those who dropped out, female infants with a good prognosis and 25–29 weeks gestation stopped coming to the hospital for follow-up checkups. Furthermore, information regarding feeding method and nutritional management during the postnatal period was not evaluated, and our study protocol did not include potential confounders, such as maternal age, complications during pregnancy, and parents’ socioeconomic status.

## 5. Conclusions

In conclusion, a younger gestational age had a greater influence on anthropometric parameters in males up to 6 years of age. A hierarchical analysis for GA and sex is needed to evaluate growth trajectory in VLBW, including periviable births.

## Figures and Tables

**Figure 1 jcm-11-01418-f001:**
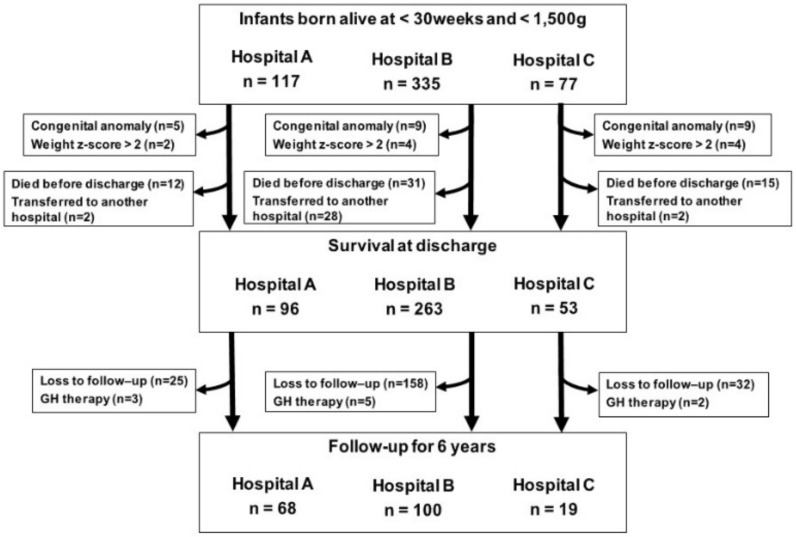
Flow chart of the study population. During the study period, a total of 529 VLBW infants with GA < 30 weeks were born in the NICU of three hospital, and 502 infants met the inclusion criteria at birth. We analyzed anthropometric data from 187 children up to 6 years corrected age, 44 were categorized into the 22–24 weeks group, 143 in the 25–29 weeks group.

**Figure 2 jcm-11-01418-f002:**
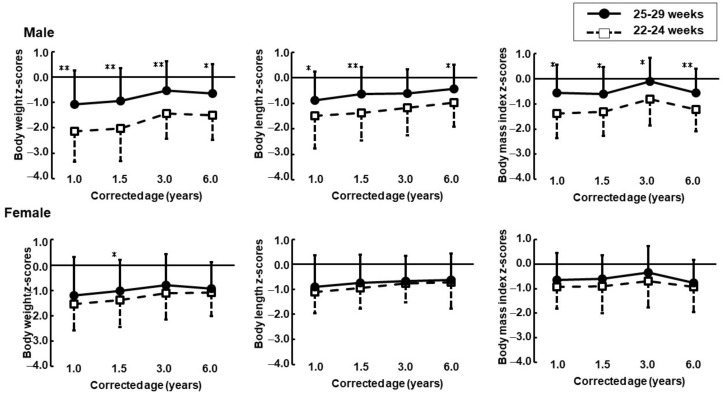
Z-scores trajectories (upper: male, lower: female) at 1, 1.5, 3 and 6 years corrected age in children born at less than 30 weeks of gestation. Solid lines: the 25–29 weeks groups, dotted lines: the 22–24 weeks groups. * *p* < 0.05, ** *p* < 0.01, compared with the 22–24 weeks group.

**Table 1 jcm-11-01418-t001:** Anthropometric indices of children from birth to 6 years of age.

		Male		Female	
		22–24 Weeks	25–29 Weeks	22–24 Weeks	25–29 Weeks
	*n*	16	84	28	59
At birth	Gestational age (weeks)	24.2 ± 0.6	27.6 ± 1.5 **	24.0 ± 0.6	27.7 ± 1.5 **
	BW (g)	635.4 ± 127.3	936.6 ± 250.7 **	605.2 ± 93.0	878.3 ± 268.4 **
	Z-scores for BW ^1^	−0.4 ± 1.2	−1.0 ± 1.2	−0.0 ± 0.8	−1.2 ± 1.3 **
	BL (cm)	29.7 ± 2.2	34.6 ± 3.0 **	29.7 ± 1.7	34.0 ± 3.7 **
	Z-scores for BL ^1^	−0.3 ± 0.9	−0.4 ± 1.1	−0.2 ± 0.7	−0.8 ± 1.2 **
	Ponderal index (g/cm^3^)	2.4 ± 0.2	2.2 ± 0.2 **	2.3 ± 0.2	2.2 ± 0.2 *
1-year CA	CA (mo)	12.3 ± 1.1	12.6 ± 1.7	11.7 ± 1.4	12.4 ± 1.8
	BW (kg)	7.6 ± 0.9	8.6 ± 1.2 **	7.5 ± 0.8	7.9 ± 1.2
	Z-scores for BW ^2^	−2.1 ± 1.2	−1.1 ± 1.3 *	−1.5 ± 1.0	−1.2 ± 1.4
	BL (cm)	71.4 ± 3.6	73.2 ± 3.4	70.2 ± 2.5	71.5 ± 3.8
	Z-scores for BL ^2^	−1.5 ± 1.3	−0.9 ± 1.1 *	−1.1 ± 0.8	−0.9 ± 1.3 *
	BMI (kg/m^2^)	14.9 ± 1.2	15.9 ± 1.4	15.1 ± 1.2	15.4 ± 1.4
	Z-scores for BMI ^2^	−1.4 ± 1.0	−0.6 ± 1.1 *	−0.9 ± 0.9	−0.7 ± 1.1
1.5-year CA	CA (mo)	17.8 ± 1.4	18.0 ± 1.7 *	17.6 ± 2.1	18.0 ± 1.8
	BW (kg)	8.6 ± 1.1	9.6 ± 1.3 **	8.6 ± 1.0	9.0 ± 1.1 *
	Z-scores for BW ^2^	−2.0 ± 1.3	−0.9 ± 1.3 **	−1.4 ± 1.1	−1.0 ± 1.2 *
	BL (cm)	76.5 ± 3.6	78.7 ± 3.3	76.3 ± 2.8	77.3 ± 3.5
	Z-scores for BL ^2^	−1.4 ± 1.0	−0.6 ± 1.1 **	−1.0 ± 0.8	−0.8 ± 1.1 *
	BMI (kg/m^2^)	14.6 ± 1.1	15.4 ± 1.3 *	14.7 ± 1.3	15.0 ± 1.2
	Z-scores for BMI ^2^	−1.3 ± 1.0	−0.6 ± 1.1 *	−0.9 ± 1.1	−0.6 ± 1.0
3-year CA	CA (mo)	33.4 ± 1.5	34.5 ± 2.1	33.7 ± 2.5	34.1 ± 1.6
	BW (kg)	11.3 ± 1.1	12.7 ± 1.7 **	11.4 ± 1.3	11.9 ± 1.6
	Z-scores for BW ^2^	−1.4 ± 1.0	−0.5 ± 1.2 **	−1.1 ± 1.0	−0.8 ± 1.2
	BL (cm)	87.8 ± 3.7	90.3 ± 3.7 **	88.2 ± 3.0	88.8 ± 3.5
	Z-scores for BL ^2^	−1.2 ± 1.1	−0.6 ± 0.9	−0.8 ± 0.8	−0.7 ± 1.0
	BMI (kg/m^2^)	14.7 ± 1.1	15.5 ± 1.3 *	14.6 ± 1.3	15.0 ± 1.3
	Z-scores for BMI ^2^	−0.8 ± 1.0	−0.1 ± 1.2 *	−0.7 ± 1.1	−0.3 ± 1.1
6-year CA	CA (mo)	70.3 ± 3.3	69.8 ± 4.3	70.7 ± 2.4	70.1 ± 3.4
	BW (kg)	16.3 ± 1.7	18.1 ± 2.8 *	16.9 ± 2.1	17.2 ± 2.4
	Z-scores for BW ^2^	−1.5 ± 1.0	−0.6 ± 1.1 *	−1.1 ± 0.9	−0.9 ± 1.0
	BL (cm)	107.7 ± 4.1	110.0 ± 4.9 *	108.7 ± 5.1	108.8 ± 4.8
	Z-scores for BL ^2^	−1.0 ± 0.9	−0.4 ± 0.9 *	−0.7 ± 1.0	−0.6 ± 1.1
	BMI (kg/m^2^)	14.0 ± 1.0	14.9 ± 1.5 **	14.2 ± 1.3	14.4 ± 1.2
	Z-scores for BMI ^2^	−1.2 ± 0.9	−0.6 ± 1.2 **	−0.9 ± 1.0	−0.8 ± 0.9

Data are presented as the mean ± standard deviation. HC, head circumference; BMI, body mass index; CA, corrected age. * *p* < 0.05, ** *p* < 0.01, compared with the 22–24 weeks group. Data are presented as means ± standard deviation (SD). Z-scores ^1^ were calculated based on Japanese neonatal anthropometric charts for gestational age at birth [12]. Z-scores ^2^ were calculated based on a Japanese growth curve [13,14].

**Table 2 jcm-11-01418-t002:** Linear regression analysis to determine the association between body weight z-scores and the presence of small for gestational age and two gestational age groups.

	Male			Female		
	Coefficient	*p*	95% CI	Coefficient	*p*	95% CI
CA 1 year						
Small for gestational age	1.033	0.000	0.469–1.597	1.128	0.002	0.441–1.814
Gestational age (group)	1.358	0.000	0.663–2.054	0.737	0.030	0.073–1.400
Constant	−3.113		−3.927–−2.299	−2.624		−3.458–−1.789
CA 1.5 year						
Small for gestational age	1.134	0.000	0.605–1.664	0.956	0.001	0.386–1.526
Gestational age (group)	1.410	0.000	0.748–2.072	0.695	0.012	0.158–1.233
Constant	−3.089		−3.861–−2.317	−2.312		−3.000–−1.624
CA 3 year						
Small for gestational age	0.952	0.000	0.483–1.421	0.736	0.014	0.153–1.319
Gestational age (group)	1.157	0.000	0.571–1.743	0.574	0.043	0.0185–1.1302
Constant	−2.319		−3.003–−1.636	−1.824		−2.531–1.116
CA 6 year						
Small for gestational age	0.890	0.000	0.424–1.357	0.479	0.071	−0.042–0.999
Gestational age (group)	1.113	0.000	0.530–1.695	0.320	0.198	−0.171–0.812
Constant	−2.335		−3.014–−1.655	−1.558		−2.186–0.930

**Table 3 jcm-11-01418-t003:** Linear regression analysis to determine the association between body length z-scores and the presence of small for gestational age and two gestational age groups.

	Male			Female		
	Coefficient	*p*	95% CI	Coefficient	*p*	95% CI
CA 1 year						
Small for gestational age	0.705	0.006	0.204–1.207	1.038	0.000	0.478–1.597
Gestational age (group)	0.797	0.012	0.178–1.416	0.574	0.038	0.034–1.114
Constant	−2.146		−2.870–−1.423	−2.112		−2.792–−1.433
CA 1.5 year						
Small for gestational age	0.619	0.009	0.158–1.079	0.809	0.002	0.297–1.321
Gestational age (group)	0.915	0.002	0.339–1.490	0.475	0.054	−0.008–0.958
Constant	−1.966		−2.637–−1.296	−1.732		−2.350–−1.114
CA 3 year						
Small for gestational age	0.433	0.045	0.010–0.857	0.548	0.024	0.074–1.022
Gestational age (group)	0.680	0.012	0.151–1.209	0.291	0.204	−0.161–0.743
Constant	−1.587		−2.204–−0.971	−1.300		−1.874–−0.724
CA 6 year						
Small for gestational age	0.488	0.020	0.078–0.898	0.392	0.153	−0.148–0.932
Gestational age (group)	0.678	0.010	0.166–1.191	0.232	0.368	−0.278–0.741
Constant	−1.437		−2.035–−0.840	−1.097		−1.749–−0.445

**Table 4 jcm-11-01418-t004:** Linear regression analysis to determine the association between body mass index z-scores and the presence of small for gestational age and two gestational age groups.

	Male			Female		
	Coefficient	*p*	95% CI	Coefficient	*p*	95% CI
CA 1 year						
Small for gestational age	0.716	0.003	0.246–1.186	0.443	0.109	−0.101–0.988
Gestational age (group)	1.033	0.001	0.453–1.612	0.426	0.111	−0.100–0.951
Constant	−2.064		−2.742–−1.385	−1.353		−2.014–0.692
CA 1.5 year						
Small for gestational age	0.867	0.000	0.416–1.317	0.431	0.103	−0.089–0.951
Gestational age (group)	0.943	0.001	0.380–1.507	0.435	0.082	−0.056–0.925
Constant	−2.121		−2.778–−1.464	−1.311		−1.938–−0.683
CA 3 year						
Small for gestational age	0.943	0.000	0.470–1.416	0.499	0.075	−0.052–1.050
Gestational age (group)	0.968	0.002	0.378–1.559	0.525	0.050	−0.000–1.051
Constant	−1.693		−2.381–−1.005	−1.179		−1.847–−0.510
CA 6 year						
Small for gestational age	0.903	0.001	0.401–1.405	0.304	0.224	−0.189–0.798
Gestational age (group)	0.926	0.004	0.299–1.554	0.268	0.257	−0.198–0.734
Constant	−2.077		−2.809–−1.345	−1.219		−1.815–−0.623

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
