# Peer review of "Growth Trajectories during the First 6 Years in Survivors Born at Less Than 25 Weeks of Gestation Compared with Those between 25 and 29 Weeks"

_jcm, 2022, doi:10.3390/jcm11051418_

Round 1

Reviewer 1 Report

I think this is a good study. 

My concern is the dropout rate as the authors have stated.  Was there a difference in gender with respect to dropout? 

Comment: the greatest  morbidity is in chronic lung disease.  Is there any difference according to growth?  

Author Response

We have revised the manuscript to address your comments. We found your comments helpful, and provided a useful perspective on our work. 

  1. My concern is the dropout rate as the authors have stated. Was there a difference in gender with respect to dropout?

Response 1: In 22-24 wks group, the rate of dropout is 44.8% (13/29) in males and 31.7% (13/41) in females. In 25-29 wks group, the rate of dropout is 54.6% (101/185) in males and 59.9% (88/147) in females. There was no statistical difference in gender analyzed by Chi-square test.

  1. The greatest morbidity is in chronic lung disease. Is there any difference according to growth?

Response 2: Although we did not compare growth trajectories according to the severity of CLD, we examined longitudinal anthropometric z-scores and the presence of SGA and youngest gestational age by linear regression analysis. We think it may not influence our results because important risk factors of CLD are also prematurity (younger gestational age) and SGA.

Reviewer 2 Report

The authors presented an important study in which the growth projections of extremely premature infants were reported. Such studies provide valuable information for monitoring the growth of these vulnerable babies with increasing survival rates. However, as stated by the authors, the retrospective nature of the study and the high number of infants out of follow-up are important limitations of the study.

In the discussion section, the authors discussed literature information rather than their own data. It would be valuable for them to provide further discussion and literature comments on their data.

Body weights (BW) in Table 1 should be corrected as "kg" for corrected ages (CA).

Author Response

We have revised the manuscript to address your comments. We found your comments helpful, and provided a useful perspective on our work.

  1. In the discussion section, the authors discussed literature information rather than their own data. It would be valuable for them to provide further discussion and literature comments on their data.

Response 1: In accordance with your suggestion, we added the following sentences in the Discussion section “Our previous study demonstrated that FGR affects the z-scores of BW and BMI in males and of BW and BL in girls during 6 years among same population assessed by ANOVA[15]. However, our following study showed the importance of GA in anthropometric parameters using larger cohort with 8838 eligible samples[16]…. Therefore, we analyzed both FGR and GA in this study. We found that FGR (nearly equal SGA) was associated with anthropometric z-scores at 6 years CA in males but not in females assessed by linear regression analysis.”.

  1. Body weights (BW) in Table 1 should be corrected as "kg" for corrected ages (CA).

Response 2: We corrected the unit of body weight in Table1.

Reviewer 3 Report

Thank you very much for your work,

It would be good if you could improve the introduction with more information and with more significant data.

  • What are the recent advances in neonatal intensive care?
  • What types of neurodevelopmental impairment could be obtained Infants born at < 25 weeks ?
  • The author must include enough information to understand the objective, with the information that they include in the introduction is impossible to understand the purpose. 
  • The information would be less global and to be directed to the objective not to the sample to study.
  • I consider the next steps (from methods and conclusions are good and they can be understood by readers)

Author Response

We have revised the manuscript to address your comments. We found your comments helpful, and provided a useful perspective on our work.

  1. It would be good if you could improve the introduction with more information and with more significant data.

Response 1: According to your suggestion, we added the following sentences in the Introduction section “Although nutritional strategies are advancing, including higher intravenous amino acids and lipid administration, and earlier achievement of full enteral feeding, postnatal (extreuterine) growth failure in VLBW infants has been a significant problem[9]. Takayanagi et al. reported that the prevalence of extrauterine growth restriction was 47.5% among 322 VLBW infants and it was significantly associated with short stature and thinness at 6 years[10]“.

  1. What are the recent advances in neonatal intensive care?

Response 2: In accordance with your suggestion, we added the following sentences in the Introduction section “Circulatory management guided by neonatologist-performed echocardiography and nutritional management promoted by early enteral feeding contributed to decreased the incidence of intraventricular hemorrhage (IVH) and necrotizing enterocolitis (NEC) in Japan[2]”.

  1. What types of neurodevelopmental impairment could be obtained infants born at < 25 weeks ?

Response 3: Most of the eligible infants were performed Bayley’s scales of infant development at 1.5 and 3 years CA. However, we could not include detailed neurodevelopmental outcome in this study. 

  1. The author must include enough information to understand the objective, with the information that they include in the introduction is impossible to understand the purpose.

Response 4:  In accordance with your suggestion, we corrected the sentences in the Introduction section.

Reviewer 4 Report

The study is a comparative one between 2 categories of gestational age, not as title present below 25 weeks of gestation.

The title  has to be reformulated accordingly as the study describe 2 category of patients.

The ponderal index formula -row 69- is wrong expressed .

Row 72 -I suggest growing charts instead growing curves

In the results section were analyzed some complication of which impact are not discussed on growth and antropometrics of patients from the study.

This has to be considered and discussed or table with data's removed.

Author Response

We have revised the manuscript to address your comments. We found your comments helpful, and provided a useful perspective on our work.

  1. The study is a comparative one between 2 categories of gestational age, not as title present below 25 weeks of gestation. The title has to be reformulated accordingly as the study describe 2 category of patients.

Response 1: In accordance with your suggestion, we changed the title of this study to “Growth trajectories during the first 6 years in survivors born at less than 25 weeks compared with older gestational ages”.

2.The ponderal index formula -row 69- is wrong expressed

Response 2: In accordance with your suggestion, we corrected the ponderal index (also Table 1).

3. Row 72 -I suggest growing charts instead growing curves

Response 3: In accordance with your suggestion, we changed to “growing charts”.

4. In the results section were analyzed some complications of which impact are not discussed on growth and anthropometrics of patients from the study. This has to be considered and discussed or table with data's removed.

Response 3: In accordance with your suggestion we omitted the Table about clinical complication. We could not analyze the influence of clinical complication on growth trajectories in this study. It is difficult for us to calculate statistics including clinical complications for covariates because of small samples.